# An In Vitro and In Vivo Assessment of Antitumor Activity of Extracts Derived from Three Well-Known Plant Species

**DOI:** 10.3390/plants12091840

**Published:** 2023-04-29

**Authors:** Octavia Gligor, Simona Clichici, Remus Moldovan, Nicoleta Decea, Ana-Maria Vlase, Ionel Fizeșan, Anca Pop, Piroska Virag, Gabriela Adriana Filip, Laurian Vlase, Gianina Crișan

**Affiliations:** 1Department of Pharmaceutical Botany, Iuliu Hatieganu University of Medicine and Pharmacy, 8 Victor Babes Street, 400347 Cluj-Napoca, Romania; gligor.octavia@umfcluj.ro (O.G.);; 2Department of Physiology, Iuliu Hatieganu University of Medicine and Pharmacy, 8 Victor Babes Street, 400347 Cluj-Napoca, Romania; 3Department of Toxicology, Faculty of Pharmacy, Iuliu Hatieganu University of Medicine and Pharmacy, 8 Victor Babes Street, 400347 Cluj-Napoca, Romania; 4Department of Radiobiology and Tumor Biology, Oncology Institute “Prof. Dr. Ion Chiricuță”, 34-36 Republicii Street, 400015 Cluj-Napoca, Romania; 5Department of Pharmaceutical Technology and Biopharmaceutics, University of Medicine and Pharmacy, 8 Victor Babes Street, 400347 Cluj-Napoca, Romania; laurian.vlase@umfcluj.ro

**Keywords:** antitumor activity, antioxidant activity, *Xanthium spinosum*, *Trifolium pratense*, green coffee beans, caffeoylquinic acid, isoflavones, xanthanolides

## Abstract

One of the objectives of this study consists of the assessment of the antitumor activity of several extracts from three selected plant species: *Xanthium spinosum* L., *Trifolium pratense* L., and *Coffea arabica* L. and also a comparative study of this biological activity, with the aim of establishing a superior herbal extract for antitumor benefits. The phytochemical profile of the extracts was established by HPLC-MS analysis. Further, the selected extracts were screened in vitro for their antitumor activity and antioxidant potential on two cancer cell lines: A549—human lung adenocarcinoma and T47D-KBluc—human breast carcinoma and on normal cells. One extract per plant was selected for in vivo assessment of antitumor activity in an Ehrlich ascites mouse model. The extracts presented high content of antitumor compounds such as caffeoylquinic acids in the case of *X. spinosum* L. (7.22 µg/mL—xanthatin, 4.611 µg/mL—4-O-caffeoylquinic acid) and green coffee beans (10.008 µg/mL—cafestol, 265.507 µg/mL—4-O-caffeoylquinic acid), as well as isoflavones in the case of *T. pratense* L. (6806.60 ng/mL—ononin, 102.78 µg/mL—biochanin A). Concerning the in vitro results, the *X. spinosum* L. extracts presented the strongest anticancerous and antioxidant effects. In vivo, ascites cell viability decreased after *T. pratense* L. and green coffee bean extracts administration, whereas the oxidative stress reduction potential was important in tumor samples after *T. pratense* L. Cell viability was also decreased after administration of cyclophosphamide associated with *X. spinosum* L. and *T. pratense* L. extracts, respectively. These results suggested that *T. pratense* L. or *X. spinosum* L. extracts in combination with chemotherapy can induce lipid peroxidation in tumor cells and decrease the tumor viability especially, *T. pratense* L. extract.

## 1. Introduction

Cancer represents a disease in which certain pathophysiological alterations occur in the process of cell division. This pathogenesis consists of various types of tumors. Though many advances have been made concerning diagnosing and treating this dreadful disease, new information regarding phenotypic plasticity and disrupted differentiation have been recently established as new hallmarks of cancer [1]. According to recent statistics, more than 19.3 million cancer cases were diagnosed and reported worldwide [2]. Phytotherapy has been proposed as an adjuvant treatment or alternative to chemotherapy, radiation therapy, immunotherapy, surgery, and many other forms of approach to treating cancer. Unfortunately, these methods have been known to cause grave side effects as well as chemoresistance. Clinically tested and familiar plant-derived bioactive compounds are vinblastine, vincristine, podophyllotoxin, paclitaxel, and camptothecin [3].

The antiproliferative activity of polyphenolic compounds is widely known throughout the current scientific literature. It occurs as a result of their antioxidant properties, ROS scavenging, chelating transition elements, and regulating the activity and synthesis of antioxidant enzymes. Examples of bioactive compounds pertaining to this class with potent antitumor activity are epigallocatechin-3-gallate, trans-resveratrol, quercetin, and curcumin [4]. Other plant-derived bioactive compounds such as acetogenins and annomuricin from *Annona muricata* L. have demonstrated in vitro cytotoxicity or emodin, an anthraquinoid compound from *Aloe vera* L., also showed in vivo cytotoxicity [3].

The plant genus *Xanthium* L. is well known for its species richness in polyphenols, namely caffeoylquinic acids (CQA). Chlorogenic represent a class of esters formed between quinic and hydroxycinnamic acids, such as caffeic, ferulic, or *p*-coumaric acid. Principal classes of caffeoylquinic acids comprise CQA, dicaffeoylquinic acids (diCQA), and feruoylquinic acids. These compounds have been reported to exhibit numerous biological activities, some of which include antioxidant, antibacterial, anti-inflammatory, and neuroprotective activities as well as antitumor effects [5]. For practical reasons, this paper focuses on several isomers of the chemical class of CQA, such as 4-O-caffeoylquinic (4-CQA), 3,4-, 3,5-, and 4,5-dicaffeoylquinic acids (diCQA). The genus’ anti-cancer potential is also due to an innately secreted class of sesquiterpene lactones, the xanthanolides. Although phytotoxic in their primary nature, xanthanolides have been demonstrated to also possess anti-inflammatory, antimicrobial, and even anti-tumor activities [6,7,8]. Xanthatin is one of the main bioactive components of *Xanthium* extracts proven to inhibit tumor growth and cell proliferation by proposed mechanisms including inhibition of the mitotic spindle assembly, inducing G2-M arrest and impairment of anaphase entrance [9,10]. For practical reasons, this study focuses on xanthatin and its isomer, 8-epi-xanthatin. *Xanthium spinosum* L. was selected for the study due to the lack of research concerning it. *Trifolium pratense* L., or red clover by its common name, is widely known for its isoflavonic compounds such as formononetin, biochanin A, ononin, daidzein, and genistein. Not only being considered phytoestrogens with estrogenic, anti-inflammatory, cardioprotective, and neuroprotective activities, these compounds have also displayed antitumor effects as well [11,12,13,14,15,16,17]. Regarding their antitumor activity, the three isoflavonic compounds, biochanin A, formononetin, and ononin, have been reported to promote apoptosis, inhibit cell proliferation and differentiation, and other cell cycle phases, as well as inhibit cellular oxidative stress [18,19,20,21]. Coffee has garnered the attention of researchers as its consumption has been associated with a decreased risk of the development of various types of cancer [22,23,24]. The polyphenolic compounds of coffee beans, especially the unprocessed green coffee beans from *Coffea arabica* L. (also named Arabica), such as chlorogenic acids and CQA have been reported to inhibit cancer cell proliferation and apoptosis [25,26]. These compounds are found in larger amounts in unroasted, green coffee beans, as opposed to the roasted plant material [27,28,29]. In addition, the diterpenoid compounds, found in the lipidic fraction of green coffee beans, such as kahweol, for instance, have been reported to reduce cyclin D1 degradation, thus inhibiting the proliferation of human colorectal cancer cells [27,30].

The purpose of this study was to assess the antitumor activity of several extracts from the three plant species (mentioned above): *Xanthium spinosum* L., *Trifolium pratense* L., and *Coffea arabica* L. In addition, the authors have accomplished a comparative study of this biological activity for the three plant species, with the aim of establishing the superior herbal extract, in regard to the antitumor benefits.

Phytochemical characterization of the extracts was first performed in order to identify and quantify the antitumor bioactive compounds. Six extracts were studied in vitro for their anticancerous and antioxidant potentials. Based on the results of the phytochemical profiling and the results from the in vitro studies, only three extracts were further subjected to the in vivo assessment of the antitumor activity.

## 2. Results

### 2.1. HPLC-MS Analysis of the Extracts

The HPLC-MS results of the initially selected extracts are presented in the following sections. The nomenclature of the samples used in this section, only, can be found in previously published works on the selected plant species [31,32,33].

#### 2.1.1. Analysis of Antitumor Compounds from the *Xanthium* Extracts

##### Xanthanolide Sesquiterpene Lactones

Two xanthanolides specific to the genus *Xanthium*, partially responsible for its antitumor activity, were quantified—xanthatin and its isomer—8-epi-xanthatin. Results are shown in Table 1. All extraction methods were able to yield the two compounds. For xanthatin, SE and UAE (30 min and 40 °C) yielded almost identical concentrations. UAE was the sole extraction method able to obtain high levels of the isomer 8-epi-xanthatin. The combination of TBE and UAE managed to attain the higher concentration levels for both compounds.

##### Caffeoylquinic Compounds

Four caffeoylquinic acids presenting antitumor activity, often found in the genus *Xanthium*, were also quantified in the selected extracts: 4-O-caffeoylquinic (4-CQA), 3,4-, 3,5-, and 4,5-dicaffeoylquinic acids (diCQA). Results are displayed in Table 1.

The highest concentration of 4-CQA was attained by SE. UAE values were comparable to those resulting from maceration, with TBE yielding the lowest levels of this compound. Results for the three diCQA were comparable to one another, as they are isomers, with few differences between the extraction methods. Respectively, UAE conditions of 30 min and 40 °C were the most favorable, closely followed by the UTE sample, and secondly by the conditions of 20 min and 50 °C. SE was the only classical method able to achieve results comparable to the lower values accomplished by UAE.

#### 2.1.2. Evaluation of Antitumor Compounds from the Trifolium Extracts

Results for this section are shown in Table 2. Three antitumor compounds were investigated: formononetin, ononin, and biochanin A. The highest concentrations were found for formononetin, followed by ononin, and biochanin A. For formononetin and biochanin A, results of the UAE extracts at 30 min, 50 °C were comparable to those of the 60 min SE extracts. In the case of ononin, only SE at 60 min time reached the highest value.

#### 2.1.3. Analysis of Antitumor Compounds from the Green Coffee Bean Extracts

##### Caffeoylquinic Acids

The samples were screened for four caffeoylquinic compounds: 4-CQA, 3,4-, 3,5-, and 4,5-diCQA. Results are shown in Table 3. The levels of these compounds were similar to one another, due to their isomerism. High yields were observed especially in the 60 min SE samples and TBE samples, namely, four cycles of 5 min extraction time and 4000 rpm and 6000 rpm, respectively. Maceration, UAE, and UTE reached only medium to high levels of caffeoylquinic acids. A progressive increase in yields along with the increase in parameter values could be observed concerning all extraction methods, for all four compounds. This suggests an improvement in compound release due to the increase in temperature, time, and speed, respectively.

##### Diterpenoid Compounds

The samples were also screened for two diterpenoid compounds of which present reports concerning their antitumor activity were found, that is, kahweol and cafestol. Results for these compounds can also be found in Table 3. Having chemical structures related to one another, these compounds exhibited similar behaviors in terms of extraction yields. Thus, the highest yields were attained for both kahweol and cafestol by UAE, specifically the 30 min and 40 °C sample, followed by other UAE samples with minor differences between each concentration value. Other extraction methods enabled only low to medium yields for these compounds. All extraction methods obtained higher diterpenoid yields as temperature and time values increased. For UAE as well as SE, a further increase in extraction time and temperature led to an abrupt decrease in diterpenoid yields due to possible degradation of the compounds as a consequence of prolonged exposure.

### 2.2. Evaluation of Biological Activities in Cell Culture for the X. spinosum L. Extracts

Once the phytochemical profile of the extracts was determined, two samples from each species were selected for further determination of in vitro biological activities. The selection was based on the number of identified antitumor compounds and the concentration levels of said compounds. Thus, for *X. spinosum* L., the extracts S60 (Soxhlet extraction, 60 min extraction time) and U34 (ultrasound-assisted extraction (UAE) at 30 °C extraction temperature, and 40 min extraction time) were selected. For *T. pratense* L., extracts S60 (Soxhlet extraction, 60 min extraction time) and U35 (UAE at 30 °C extraction temperature, 50 min extraction time). For green coffee beans, the extracts S60 (Soxhlet extraction, 60 min extraction time), and U35 (UAE at 30 °C extraction temperature, 50 min extraction time). Further, in order to promote the legibility of the study, the extracts were renamed in the sections discussing biological activities (see Table 4).

#### 2.2.1. Evaluation of Cell Viability

The anticancerous effects of *X. spinosum* L. extracts were evaluated in parallel on cancerous and normal human cell lines (Figure 1). For the cancerous phenotype, human lung adenocarcinoma (A549) cells and human breast carcinoma (T47D-KBluc) were selected. As a normal cell line, human fibroblast cells (BJ) isolated from the foreskin were used. The evaluation was done after a 24 h and a 48 h incubation with the extracts at doses ranging from 50 to 1000 µg/mL.

Both extracts induced cytotoxic effects on all the cell lines used, with higher cytotoxicity being observed on the cancerous cell phenotypes. The cytotoxic effect was statistically significant on the cancerous cells starting from the lowest exposure dose of 50 µg/mL. Starting from this dose, a dose-dependent increase in cytotoxicity was observed for both extracts on the A549 and T47D-KBluc cells. Higher cytotoxicity was observed in the case of breast cancer cells, at concentrations above 300 (XS) and 400 (XU) µg/mL and an exposure of 24 h, the extracts reducing the cellular viability by almost 100%. This total cellular death was present only at the highest two concentrations of 800 and 1000 µg/mL in the case of A549 cells. Exposure of the cancerous cells to the extracts for 48 h induced a more pronounced cytotoxic effect.

Conversely, in the case of normal cells, an increase in the measured viability was recorded at low and intermediary doses. This hormetic response was more pronounced for the XU extract and after an exposure of 48 h. The cellular viability of BJ cells was increased up to 105 and 125% after a 24 h exposure to XS and XU, respectively. After a 48 h exposure to the XS and XU, the viability of BJ cells increased up to a value of 135 and 140%.

The hormetic effect observed could be related to the presence of CQA in both extracts, recent studies indicate that these phytocompounds increase intracellular adenosine triphosphate (ATP) levels in cells and thus the metabolic conversion of resazurin to resorufin [34]. Structure–activity relationship studies indicate that the activity is proportional to the number of caffeoyl moieties in the quinic acid, and explain the higher cellular viability measured in the case of XU which is more abundant in dicaffeoylquinic acids than the XS extract (Table 1) [35]. Even though low and intermediary doses increased the measured cellular viability, high doses of extracts (>600 µg/mL) induced strong cytotoxicity in the normal cells. At the highest tested dose of 1000 µg/mL, both extracts reduced the cellular viability to almost zero at both times of exposure.

For the potency evaluation of the *X. spinosum* L. extracts, 4-parameter logistic curves with the cellular viability as a function of the exposure doses were plotted, and the IC_50_ (half maximal inhibitory concentration) values were extrapolated (Table 5). The calculated IC_50_ values mirrored the previous observations, with lower values being obtained for the cancerous cells than for normal fibroblasts.

The IC_50_ values for the T47D–Kbluc cells were approximately half of the values of A549 cells and a third of the values of BJ. Based on IC_50_ values, a more pronounced cytotoxic effect was observed for XS on the cancerous cells, whereas XU displayed higher cytotoxicity on the normal cells. Interestingly, the same XU extract that displays slightly higher cytotoxicity on the normal cells, induces a higher increase in cellular viability at intermediary doses. A three-way ANOVA with the cell type, extract type, and exposure dose as variables and the measured cellular viability as a response was performed to evaluate the differences between the conditions. The results of the statistical test indicated that all the variables significantly influenced the observed viability, the cell type, and the dose being the main contributors to the cellular viability observed.

#### 2.2.2. Evaluation of Cell Viability

The antioxidant potentials of the *X. spinosum* L. extracts (XS and XU) were evaluated at three concentrations (100, 200, and 300 µg/mL) that did not induce cytotoxicity on the normal human fibroblast cells (Figure 2). The evaluation was done using the DCFH-DA assay, as previously described, in the presence or the absence of a pro-oxidant (H_2_O_2_) [36]. Exposure alone for 24 h with both extracts decreased in a dose-dependent manner the quantity of ROS. The antioxidant properties were more prominent for the XS extract, at the highest tested dose of 300 µg/mL decreasing ROS more than the positive control (NAC). For XU extract, in non-stimulated conditions, antioxidant effects similar to those observed for NAC were achieved at a dose of 300 µg/mL. Treatment with H_2_O_2_ increased by approximately 250% ROS quantities, an increase that was partially mitigated by NAC. Similar to NAC, both extracts partially decreased ROS in stimulated conditions, the statistical significance is observed from the lowest dose of 100 µg/mL for both extracts. Similar to the observations made in non-stimulated conditions, XS had a better antioxidant profile, with the highest tested dose decreasing ROS more than the positive control. XU extract had the same capacity as NAC in abrogating ROS at the maximum concentration.

### 2.3. Evaluation of Biological Activities in Cell Culture for the T. pratense L. Extracts

#### 2.3.1. Evaluation of Cell Viability

In the case of both *T. pratense* L. extracts, higher cytotoxicity, which was dose-dependent, was observed in cancerous cells (Figure 3, Table 6). After a 24 h exposure to the extracts, the viability of A549 and T47D-KBluc cells statistically decreased from the doses of 300 and 150 µg/mL, respectively. The higher sensitivity of T47D-KBluc cells to the *T. pratense* L. extracts was also observed at higher doses, at the maximum concentration tested of 1000 µg/mL both extracts induced an almost 100% cytotoxicity.

In the case of A549 cells, a 24 h exposure to both extracts at the highest dose tested decreased the cellular viability by approximately 65%. Interestingly, an increase in viability was observed for normal cells at low and intermediary doses (Figure 4). The obtained effect could be related to a hormetic response, non-toxic doses of *T. pratense* L. extracts increasing the metabolism of the cells and thus the analytical signal measured by the AB assay. At these doses, an increase in the cellular viability of up to 125% was observed. At the highest tested dose, both extracts induced a cytotoxic effect in BJ cells, decreasing the cellular viability by approximately 20% (Figure 4). In comparison with an exposure of 24 h, an exposure of 48 h to the *T. pratense* L. extracts was associated with higher cellular toxicity, especially in cancerous cells. Even though *T. pratense* L. extracts have been shown to possess anticancerous activities on different types of cancerous cells, isoflavonoid compounds abundant in *T. pratense* L. extracts, such as formononetin, show increased efficiency in estrogen receptor-positive cells, thus explaining the higher sensitivity of the T47D-KBluc cells [37].

For the anticancerous potency evaluation, IC_50_ values were calculated from four-parameter logistic curves (cellular viability as a function of exposure dose), see Table 6. IC_50_ values reflected the previous observations, T47D-KBluc cells being more sensitive to the cytotoxic effects of the *T. pratense* L. extracts.

#### 2.3.2. Evaluation of the Antioxidant Potential

For the evaluation of the antioxidant effects of the *T. pratense* L. (TS and TU) extracts, three concentrations (100, 200, and 300 µg/mL) that did not induce significant cytotoxicity in BJ cells were selected. The antioxidant properties were evaluated using the DCFH-DA assay, as previously described [36]. The antioxidant potential was evaluated in stimulated/non-stimulated conditions, in the presence/absence of H_2_O_2_. Treatment alone with both extracts decreased in a dose-dependent manner to ROS (Figure 4). These effects were similar to NAC treatment, the positive control used. Exposure of cells to H_2_O_2_ induced a marked increase in ROS, the levels being approximately 250% of those observed for negative control. Pre-treatment with NAC for 24 h partially alleviated the increase in ROS induced by the H_2_O_2_. Similar to the antioxidant effects of NAC, both extracts decreased ROS at the exposure doses used (Figure 4). In the case of TS, a statistical decrease was observed starting from the dose of 200 µg/mL in non-stimulated/stimulated conditions, whereas in the case of TU, the decrease was present starting from the lowest dose evaluated of 100 µg/mL. Moreover, the antioxidant properties of TU were superior to the ones of the positive control, a statistically significant difference being observed at a dose of 300 µg/mL (Figure 4). The obtained results are congruent with the previously described antioxidant activity observed in the non-cellular assays [32].

### 2.4. Evaluation of Biological Activities in Cell Culture for the Green Coffee Bean Extracts

#### 2.4.1. Evaluation of Cell Viability

The anticancerous potential of the green coffee bean extracts (S60 and U35) was assessed as previously described for the other two species. For both extracts, higher toxicity was observed in the case of cancerous cells, whereas in the case of normal cells, no toxicity was observed. After a 24 h exposure to both extracts, the viability recorded for A549 and T47D-KBluc cells decreased by increasing the exposure dose. At the highest tested dose, the viability of A549 at 24 h post-exposure was approximately 40 and 60% for CS and CU extract, respectively. A slightly higher cytotoxicity was observed in the case of T47D-KBluc cells, and at the highest tested dose, the recorded viabilities were approximately 20% for both extracts. A similar dose-dependent cytotoxicity profile on the cancerous cell lines was observed for both *Coffea arabica* L. extracts after incubation for 48 h. However, in this case, the cytotoxic effects were more pronounced, with both extracts decreasing the cellular viability below 50% at the exposure dose of 600 µg/mL. At the highest two doses of 800 and 1000 µg/mL, exposure to the extracts induced total cell death, with the recorded viabilities being approximately 0% (Figure 5).

Conversely, to the results obtained on the cancerous phenotypes, exposure of normal cells to low and intermediary (50–400 µg/mL) concentrations of the extracts for 24 h increased the cellular viability by 10–15%. The increase in viability can be explained by the presence of CQA in the extracts, as these phytocompounds have been shown to increase the metabolic reserve and metabolic capacity of cells [34]. Recent studies indicate that this benefic activity is proportional to the number of caffeoyl moieties in the quinic acid [35]. Regarding the content of CQA in the studied extracts, the extracts are almost alike in terms of qualitative and quantitative composition (Table 7). Due to this similarity, no difference in the cellular viability of the BJ cell line was observed between the two extracts. After a 48 h incubation with the extracts, the increase in viability on the normal cells was even higher, the viability reaching 140% of the value of negative control (Figure 5).

Based on the experimental data, half maximal inhibitory concentration (IC_50_) values were extrapolated from four-parameter logistic curves of the cellular viability as a function of the exposure dose (Table 7). As to previous observations, the extracts displayed a similar potency, with slightly lower IC_50_ values being observed for the CS extract. Compared to a 24 h exposure, exposure for 48 h resulted in more pronounced toxicity in the A549 cells, the calculated IC_50_ at this time point being approximately half of the values obtained after a 24 h exposure. For T47D–Kbluc the differences between the IC_50_ after a 24 h and a 48 h exposure were not as prominent. The data were also analyzed using a Three-Way ANOVA with the measured cellular viability as a response, and the cell type, extract type, and exposure dose as variables at 24 h and 48 h exposures. In both cases, there was a statistical difference, with all variables influencing the measured viability. There was a clear difference between the normal and cancerous cell types, with the two phenotypes behaving differently. Moreover, the difference between the extracts increased at 48 h, with CS displaying higher potency.

#### 2.4.2. Evaluation of the Antioxidant Potential

The antioxidant potentials of the green coffee bean extracts were evaluated at three concentrations (100, 200, and 300 µg/mL) that did not induce cytotoxicity on the normal human fibroblast cells. The evaluation was done using the DCFH-DA assay, as previously described, in the presence or the absence of a pro-oxidant H_2_O_2_ [36]. Exposure alone for 24 h with both extracts decreased in a dose-dependent manner the quantity of ROS. For CS, a statistical significance was observed starting from the intermediary dose of 200 µg/mL, whereas for CU, the difference was observed starting from the lowest dose of 100 µg/mL. Treatment with H_2_O_2_ increased by approximately 250% ROS quantities, an increase that was partially mitigated by NAC. Similar to NAC, both extracts partially decreased ROS in stimulated conditions, the statistical significance is observed from the dose of 200 µg/mL for both extracts (Figure 6).

### 2.5. In Vivo Antitumor Activity Assessment

#### 2.5.1. Tumor Cell Viability

Cell viability is an important assessment in the evaluation of the cytotoxicity of any bioactive compound. Results are presented in Figure 7. In addition to evaluating each extract separately, as stated above, a comparison between their effects was also attempted. Thus, CS, that is, the green coffee bean extract obtained by SE at 60 min extraction time presented the lowest cell viability (*** *p* < 0.001 vs. control group), compared to CYC. This was followed by TS, that is, the *T. pratense* L. extract obtained by SE, at 60 min extraction time (** *p* < 0.01 vs. control group), in comparison to CYC. XS, that is, the *X. spinosum* L. extract obtained by SE, at 60 min extraction time did not show any significant difference versus the CYC group or control, untreated group.

#### 2.5.2. Evaluation of Oxidative Stress Markers from Ascites Samples

Redox imbalance markers determined in ascites samples can be seen in Figure 8. MDA levels increased after CYC treatment compared to controls, whereas the associations between CYC and XS or TS extracts caused a significant statistical increase in lipid peroxidation (*p* < 0.05). TS extract induced oxidative stress in ascites samples of mice close to the CYC group (*p* < 0.05), whereas the association with TS and CYC maintained the same values of MDA levels. Regarding the levels of non-endogenous antioxidants, for GSH, TS treatment and the association between CYC + CS showed comparable results, yet statistical significance was lacking. In the case of GSSG levels, there were no statistically significant differences between groups. Nevertheless, the GSH/GSSG ratio was slightly ameliorated after treatment with each extract individually. In the case of enzymatic antioxidants, CAT activity decreased after CYC treatment, whereas the extracts or extracts associated with CYC did not significantly modify the activity of the enzyme. GPx activity decreased after CYC administration, whereas the extracts elevated the enzyme activity, most notably XS and CS. Comparable results were reached by TS treatment and CYC + XS treatment. None of the data mentioned earlier presented statistical significance.

#### 2.5.3. Evaluation of Oxidative Stress Markers from Liver Samples

Figure 9 presents the results of oxidative stress from the liver samples. MDA formation slightly increased after CYC administration but was statistically insignificant, whereas CYC + XS, that is, the *X. spinosum* L. SE extract, decreased MDA formation (*p* < 0.05), suggesting its antioxidant effect. Of the non-endogenous antioxidants, liver GSH levels were slightly increased for all individual extract treatments in comparison to the CYC group, as well as for the combination treatments. However, statistical significance was not present for these results. The liver GSSG levels increased after CYC administration but unfortunately decreased after treatment with extracts as well as the combination of therapies. Nonetheless, the GSH/GSSG ratio was positively impacted by the treatments containing extracts, with TS being moderately significant (** *p* < 0.01 vs. control group). Enzymatic antioxidant activities significantly increased compared to the control group and CYC group, namely TS (## *p* < 0.01 vs. control group), CS (** *p* < 0.01 vs. control group, and ### *p* < 0.001 vs. CYC group), for CAT. For GPx, the association CYC + XS was the only significant amelioration of enzyme activity (* *p* < 0.05 vs. the control group, and # *p* < 0.05 vs. the CYC group).

## 3. Discussion

Phytochemically-wise, Varga et al. have also reported the presence of xanthatin and its isomer, 8-epi-xanthatin, as well as CQA, such as 4-CQA and 3,5-diCQA in methanolic extracts of *X. spinosum* L. aerial parts, after extraction by UAE, followed by solid-phase extraction [37]. A notable observation was that of xanthatin being present in higher levels in the S60 sample, whereas its isomer, 8-epi-xanthatin, was present in higher concentrations in the UAE samples. A possible explanation could lie in the extraction parameters themselves, namely, UAE, with milder working conditions, lower temperatures (10–30 °C), and shorter extraction times (40–50 min), could have led to the loss of the isomer in the S60 sample (extraction conditions: boiling point of the hydroalcoholic solvent of approximately 79 °C, and 60 min extraction time), but its extraction by UAE. Another possibility that could be considered is the epimerization itself of xanthatin in the conditions of the UAE. As stated above, chlorogenic acids are a phytochemical class of esters formed between quinic and *cis*- or *trans*-cinnamic acids, such as caffeic, ferulic, or *p*-coumaric acid. The main classes of chlorogenic acids are CQA, diCQA, and feruoylquinic acids. These compounds have been reported to exhibit numerous biological activities, some of which include antioxidant, antibacterial, anti-inflammatory, and antitumor activities as well as neuroprotective effects [5]. Xanthatin has also been reported to inhibit both tumor growth and cell proliferation in various types of cancer cell lines [8,9,10,38,39].

Isoflavones were richly present in the *T. pratense* L., especially for UAE and SE extracts. The lower concentrations of ononin present for all extracts, with the exception of the S60 extract, as stated above, could be due to the chemical relation between ononin and formononetin. As ononin is the glycosylated derivative of formononetin. The loss of beta-D-glucopyranoside might be caused by the prolonged extraction times combined with the elevated extraction temperatures of the UAE. Or possibly due to the shear forces of the disperser, in the case of TBE.

CQA are specific to green coffee beans and have been reported to present antitumor activity [40,41]. Guglielmetti et al. have also reported a positive correlation between time and temperature for CQA yield in coffee silverskin, extracted by several extraction methods, including UAE [42]. Bianchin et al. have also reported a positive influence of longer extraction time over kahweol and cafestol yields for roasted coffee beans, subjected to UAE. UAE was also reported to have provided comparable results to a conventional liquid/liquid extraction; a finding not applicable to the present study [43]. Moeenfard et al. have also reported the UAE efficiency over a simple solvent extraction for cafestol esters yields in samples of green and roasted coffee beans, with the main reasons being a better mass transfer, swelling and enlargement of cellular wall pores, and improved intracellular penetration of the extraction solvent [44]. Two other studies have also reported the advantage of another innovative extraction method, microwave-assisted extraction (MAE), presented in terms of diterpene content, over conventional methods such as SE and conventional heating, for green coffee beans [45,46]. The degradation of the two diterpenoid compounds by long exposure to elevated extraction temperature and time values has also been reported by Chen et al. for green coffee oil, using a combination of UAE and MAE [47].

Similar to the current results, Ly et al. reported an IC_50_ value of 81.69 μg/mL for an ethanolic extract of *X. strumarium* L. leaves on HepG2 cells after an exposure of 48 h. Furthermore, the authors reported that treatment with the extracts induced cellular apoptosis marked by shrinkage of the cellular volume and the presence of apoptotic bodies [48]. The anticancerous activity of the *X. strumarium* L. extracts was also reported on other hepatic cancerous cells (Huh-7 and Hep3B). Kim et al. evaluated the cytotoxic potential of an ethanolic extract obtained from the fruits of *X. strumarium* L. and reported IC_50_ values of around 100 μg/mL on Huh-7 and Hep3B cells after a 72 h exposure. The authors reported that apoptosis was initiated in cancerous cells after the activation of caspase-3 and PARP. Moreover, in an ex vivo model, administration of the extracts inhibited tumoral growth and induced apoptosis by suppressing the PI3K/AKT/mTOR signaling pathways in the cancerous cells [49]. Comparable IC_50_ values were also reported for other cancerous cell types, including colorectal adenocarcinomas (HT-29 and C2BBe-1), breast adenocarcinomas (MDA-MB-231 and MCF-7), and ovarian carcinomas (SKOV-3 and ES-2) after treatment with an ethanolic extract of *X. strumarium* L. leaves [50]. Higher cytotoxicity was reported in Chinese hamster ovary cells after treatment with the whole extract of the aerial parts of *X. strumarium* L. The calculated IC_50_ in this latter case was 20.5 μg/mL [51]. Although not desirable due to the environmental impact, extraction with other non-polar solvents such as chloroform could result in extracts with higher concentrations of biologically active compounds. In a recent study, Alsabah et al. reported IC_50_ values between 2 and 3 μg/mL after a 72 h exposure to a chloroform-based extract from the leaves of the same species on three different breast cancer cell lines (AMN3, AMJ13, and MCF7) [52]. Same as in our study, the authors observed that their extract exerts more cytotoxicity on cancerous cells than on normal cells [52]. Similarly, higher cytotoxicity was observed in A549, HepG2, Jurkat, and L929 cells after treatment with an ethyl acetate fraction of a methanol extract of *X. strumarium* L. seed than with the whole methanol extract. In comparison with the crude extract, the ethyl acetate fraction was enriched in more lipophilic compounds thus explaining the observed differences. Interestingly, further purification of the ethyl acetate fraction yielded subfractions that lost their anticancerous properties, reiterating that the combination of different compounds is responsible for the inhibitory effect [38]. Slightly higher cytotoxicity than of the whole crude extract of *X. strumarium* L. leaves was also for a chloroform fraction by Piloto-Perrer et al. in CT26WT colon carcinoma cells [9]. Besides the evaluation of whole crude extracts, many scientific papers tried to fractionate crude extracts and isolate compounds with antitumoral activities from different parts of *X. strumarium* L. [10,49,53,54,55,56,57]. Xanthanolide sesquiterpene lactones, including xanthatin and 8-epi-xanthatin that were also quantified in the current study, are one of the most studied phytocompounds present in the *Xanthium* genus, being partially responsible for the anticancerous properties. Compared with crude extracts, these compounds present lower IC_50_ values. Ahn et al. reported antitumor activity for xanthin and 8-epi-xanthatin on several cancerous cells, with IC_50_ values varying between 0.1 and 1.7 µg/mL, depending on the cell line employed [58]. Similarly, Ramírez-Erosa et al. reported IC_50_ values of 13.9 and 4.8 µg/mL for these compounds in MDA-MB-231 cells and IC_50_ values of 6.15 and 2.65 µg/mL in WiDr (human colon cancer) cells [59]. Based on recent research, xanthatin alters the enzymatic activity of thioredoxin reductase by targeting a selenocysteine residue. The inhibition of the enzyme is followed by the induction of apoptosis due to increased oxidative stress [60]. Other effects related to an oxidative stress imbalance such as genotoxicity and cell cycle arrest were reported as mechanisms for the induction of apoptosis and the overall anticancerous effect [51,61].

Regarding the *T. pratense* L. extracts, both displayed similar cytotoxicity patterns, the calculated IC_50_ values being almost the same in both cancerous cell types (Table 6). Congruent with these results, Karakas et al. reported IC_50_ values higher than 200 µg/mL for *T. pratense* L. extracts obtained after extraction with hexane, dichloromethane, methanol, and water. The authors evaluated the cytotoxic effect on human breast cancer cells (MCF-7) and human hepatocellular carcinoma (HepG2) after an exposure of 24 h [62]. Similarly, Khazaei et al. reported that a hydroalcoholic extract of *T. pratense* L. presented a dose and time-dependent cytotoxicity against glioblastoma (U87MG) and mammary cancer (MCF-7 and MDA-MB-231) cells [63,64]. After a 24 h exposure to U87MG cells, the reported IC_50_ value was around 400 µg/mL, close to the values presented in our study [64]. In the case of the mammary cancerous cells, the calculated IC_50_s were higher than 1000 µg/mL for both cell types after an exposure of 24 h [63]. The anticancerous potential was also reported by Zakłos-Szyda et al. that observed higher cytotoxicity on cancerous cells than in normal human umbilical vein endothelial cells [36]. In addition to the polyphenolic and flavonoid compounds that have been shown to possess anticancerous activities, biochanin A, an isoflavone present in the TS and TU extracts, has been shown to possess anticancerous properties [19]. Biochanin A manifests antineoplastic properties by inhibiting cancerous cellular growth, activating apoptosis, and blocking angiogenesis. Its activities have been demonstrated in various cell types, including bladder, pancreatic, prostate, liver, and breast cancers [19]. Moreover, formononetin and ononin, another tw4o compounds present in *T. pratense* L. extracts, have been shown to possess anticancerous activity by inducing cell apoptosis and cell cycle arrest in cancerous cells [20].

Similar to the current in vitro results for the green coffee bean extracts, Amigo-Benavent et al. reported that an exposure to 1000 µg/mL methanolic extract of green coffee beans decreased the cellular viability of A549 cells by 40 and 45% after a 2 h and 24 h incubation, respectively [26]. These results are congruent with our results. After a 24 h exposure at a dose of 1000 µg/mL, the recorded viabilities decreased by 60% for CS and 40% for CU. Different from our study, the previously mentioned study reported relatively high toxicity on normal cells, exposure to the extract at a concentration of 1000 µg/mL for 24 h decreasing the cellular viability to 22% [26]. Even though fibroblasts were used as normal cells in both studies, the different origins could explain this discrepancy, as in our case we used normal dermal fibroblasts, and in their case, the fibroblasts were isolated from colon tissue. From the phytocompounds tested, 5-CQA and 3,5-diCQA were the most potent anti-proliferative phenols, their synergistic effect, together with the potential contribution of other bioactive compounds explaining the anticancerous effects [26]. Congruent results were also reported by Priftis et al. who observed a more than 20% decrease in viability at concentrations higher than 400 µg/mL of an aqueous extract of green coffee beans on the endothelial EA.hy926 cells [65]. Interestingly, they reported much higher cytotoxicity (viability < 80%) on the myoblastic C2C12 cells, cytotoxicity being present starting with an exposure dose of 20 µg/mL. C2C12 cells were less prone to toxicity when treated with extracts from roasted coffee beans, the exposure concentrations needed to decrease the cellular viability by more than 20% being at least 20 times higher than in the case of green coffee bean extract [65]. These results could be explained by the formation of melanoidins due to Maillard reactions between sugars and amino acids that present a high molecular mass and do not pass the cellular membranes. In addition to sugars and proteins, polyphenols can be sequestrated in these structures thus affecting the bioavailability of phytocompounds [66]. In a recent paper from 2022, the anticancerous potential of green, roasted, and spent coffee aqueous and ethanolic extracts was evaluated. Similar to the obtained results, the green coffee bean extract did not induce overt cytotoxicity up to a dose of 200 µg/mL and was less toxic than their roasted counterparts [67].

Cafestol and kahweol, two coffee-specific phytocompounds present in our extracts, have been shown to possess anti-carcinogenic, anti-angiogenic, and anti-inflammatory activities [68,69,70]. Several studies indicated that cafestol and kahweol display an inhibitory activity in multiple stages of cancer development by preventing tumorigenesis, metastasis, inhibiting cell proliferation, and inducing cellular apoptosis. One of the main antiproliferative mechanisms of action for cafestol is related to the downregulation in the expression of anti-apoptotic proteins such as Bcl-2. This effect has been reported in several studies on different cancer cell types, including human renal Caki cells, human glioma U251MG cells, and human breast carcinoma MDA-MD-231 cells [71,72,73]. Similar to cafestol, the structural-related kahweol displays anticancerous activities by down-regulating the expression of anti-apoptotic proteins and by inducing the caspase 3 activation and cytochrome release from the mitochondria [74,75,76,77]. In A549 cells, Kim et al. reported that kahweol exhibited antiproliferative and pro-apoptotic effects in a dose- and time-dependent manner. Moreover, cafestol and kahweol do not induce cytotoxicity on normal cell types, suggesting a cytotoxicity selectivity towards the cancerous phenotypes [71,75,78].

The antioxidant potential of the selected extracts from the three species observed on the BJ cell line is supported by the data obtained in the abiotic DPPH, FRAP, and ABTS^+^ antioxidant assays from previous works [31,32,33]. Main groups of active principles such as phenolics (chlorogenic acid, quercetin, caffeic acid) and CQA display antioxidant properties in vivo and in vitro. These compounds can directly neutralize ROS or indirectly can induce the nuclear translocation of Nrf2 and, as a consequence, increase the synthesis of cellular antioxidants (glutathione) and the activity of detoxifying enzymes (Heme-Oxygenase 1) alleviating the insults induced by ROS. One of the phenolic compounds with the highest concentration in the green coffee bean and *X. spinosum* L. extracts, namely chlorogenic acid, is recognized for its antioxidant and anti-inflammatory potential. Several studies showed that chlorogenic acid has a protective effect on LPS-induced inflammation in hepatic stellate cells and intestinal epithelial cells [79,80]. Treatment with chlorogenic acid inhibited the production of ROS in LPS-stimulated hepatic stellate cells and decreased the nuclear translocation of the pro-inflammatory NF-κB similar to the reference antioxidant NAC [79]. In a recent study, besides the direct free radical-scavenging activity, treatment with chlorogenic acid of H_2_O_2_-stimulated cells strikingly rescued cells from oxidative insults and cellular death. The authors reported that the effect is mediated by the upregulation of antioxidant cellular defense mechanisms such as heme oxygenase-1, glutathione, thioredoxin reductase 1, and thioredoxin 1 after the nuclear translocation of Nrf2 [81]. Besides activity in vitro, chlorogenic acid has been shown to act as an oxidant in ex vivo and in vivo models [82,83,84]. CQA that were present in the green coffee bean and *X. spinosum* L. extracts are a recognized class of antioxidants, the antioxidant capacity being proportional to the number of the caffeoyl groups bound to the quinic acid [5]. The antioxidant properties of this class have been shown in cell culture, an extract of *Gymnaster koraiensis* Kitam, and one of its major components, 3,5-di-CQA, exerts protective effects on hepatic cells (HepG2) treated with the pro-oxidant tert-butyl hydroperoxide (t-BHP) [85]. Moreover, the authors reported increases in the cellular glutathione level and a decrease in BHP-induced DNA damage in cells treated with the extracts. Similar to the increase in glutathione observed in the t-BHP model, a hexane fraction of the extract mitigated the acetaminophen-induced reductions in GSH levels, protecting the cells against cellular death [85]. Similarly, other research indicated the antioxidant properties of various caffeoylquinic acids in hepatic and neuronal cells [86,87,88,89,90,91]. The antioxidant mechanism of caffeoylquinic acid compounds is similar to the mechanism of chlorogenic acid, with both phytocompounds favoring the nuclear translocation of the Nrf2 with the consequent increase in the antioxidant cellular defense mechanisms [91].

The obtained results for the in vitro antioxidant effect of *T. pratense* L. extracts are congruent with the previously described antioxidant activity observed in the non-cellular assays (Table 6) and the data present in the literature [92,93,94]. Recently Lee et al. reported the anti-inflammatory and antioxidant effects of a *T. pratense* L. extract and an anthocyanin-enriched fraction from the *T. pratense* L. extract [95]. Both extracts alleviated the increase in ROS induced by the exposure of RAW-267.4 macrophages to lipopolysaccharides. The decrease in ROS was mediated by the inhibition of gene expression and phosphorylation of NADPH oxidase 1. Moreover, the extracts exerted anti-inflammatory potential by suppressing the nuclear translocation of the NF-kB [95]. Zakłos-Szyda et al. reported the antioxidant properties of the *T. pratense* L. extracts in non-stimulated conditions and in the presence of a pro-oxidant (t-BOOH). In both cases, treatment with *T. pratense* L. extracts decreased ROS in several cell types by approximately 20–40% [36]. In addition to the polyphenolic compounds with known antioxidant properties, the isoflavones formononetin and biochanin A have been shown to act as antioxidants [96,97]. In a recent study, Sugimoto et al. have shown that formononetin decreases ROS and mitigates the cytotoxicity of H_2_O_2_ in neuronal SH-SY5Y cells by enhancing the expression of antioxidant genes [97], whereas Wu et al. reported that biochanin A attenuates the production of ROS in a dose-dependent manner in LPS-stimulated microglial cells [96]. Other isoflavonoid compounds, such as genistein and daidzein, have been shown to elicit antioxidant properties, explaining the current results [98,99,100].

There is an incongruence (especially statistical) between the data obtained from the cell cultures vs. that obtained from the in vivo experiment. One possible explanation could be due to the wide evolutive difference between the species from which the samples were taken, for instance, the cell lines were of human origin (A549—human lung adenocarcinoma, T47D-KBluc—human breast cancer), and the tested animals were Swiss albino mice. Another possible explanation could be that, as seen in the in vitro assessments, large concentrations of extracts were necessary in order to obtain an antioxidant or antitumor effect. In this case, potentially, the doses of extracts administered to the subjects of the animal model were insufficient in order to induce an appropriate, statistically significant effect. Additionally, the tumor used was an aggressive one, a metastatic ovarian tumor with poor response to different treatments. The extracts used for additive effect with CYC have demonstrated efficacy on this type of tumor. As can be noted in the liver mouse samples, the major imbalance between the levels of GSH and GSSG might be due to the polyphenolic compounds’ induction of ROS which in turn led to the induction of cell apoptosis by mitochondrial enzyme activation. As is known, the antiproliferative properties of polyphenolic compounds not only lies in their antioxidant but also in their prooxidant properties [4].

Al Kury et al. managed to demonstrate the anticancer effects and immunomodulatory activities in a mouse model for an extract of *X. spinosum* L. obtained by maceration. The authors reported an enhancement of caspase-3 activity by 2.25 folds in treated mice compared to the negative control. The authors emphasized the pharmacological activities of the flavonoid compound eupatilin, present in higher levels in an ethanol fraction of the initial extract. They also noted that the aqueous fraction of the extract, containing xanthatin in higher levels, showed the highest fold increase in caspase-3 activity compared to other extracts and fractions [101].

In a review, Sarfraz et al. observed the apoptotic effects of biochanin A by blocking pro-inflammatory factors as well as caspase activity in rat chondrocytes [19]. Kim et al. studied the antitumor effects of formononetin an athymic NU/NU nude mouse model with multiple human myeloma xenografted tumors. The authors concluded that formononetin inhibited cell viability, induced apoptosis, and also suppressed constitutive STAT3 (tyrosine residue 705 and serine residue 727) and STAT5 (tyrosine residue 694/699) activation by suppressing the upstream kinases (JAK1, JAK2, and c-Src) via increased production of ROS due to a GSH/GSSG imbalance [102]. Gong et al. were able to demonstrate the antitumor effects of ononin on BALB/c nude mice injected with A549 cells, especially in combination with paclitaxel treatment. Respectively, the combined treatment inhibited the progression of non-small-cell lung cancer (NSLC) by blocking the PI3K/Akt/mTOR pathway, thus inducing cell apoptosis and limiting cell proliferation, migration, and metastasis [103].

Coffee has garnered the attention of researchers as its consumption has been associated with a decreased risk of the development of various types of cancer. Such studies varied from observing the effect of coffee bean consumption on the risks of brain tumors, including gliomas, to liver cancer (only a correlation was noticeable), and skin cancer (in this case, in a dose dependent-manner) in several Asian populations (Japanese and Chinese Singaporean patients) [24,25,26]. The polyphenolic compounds of coffee beans, especially the unprocessed green coffee beans, such as chlorogenic acids and CQA have been reported to inhibit cancer cell proliferation in a dose dependent-manner and possibly induce cell apoptosis by inhibiting lipoxygenase activity [26,30]. As stated above, these compounds are found in larger amounts in unroasted, green coffee beans, as opposed to the roasted plant material [27,28,29]. Despite this, several cohort studies have indicated that the consumption of green and/or tea did not ameliorate the risk of urinary cancer or biliary tract cancer in human patients, particularly for male smoker patients, concerning the formerly mentioned disease. For the first study, the authors also considered the lower rates of chronic disease in the population compared to the coffee nondrinker subjects, for instance, and other socio-economic factors, which sadly affected the lifestyles and quality of life of the studied patients [69,104]. Regarding the diterpenoid compounds, Park et al. have reported kahweol reduces cyclin D1 degradation in human colorectal cancer cells. The proposed mechanism consisted of threonine-286 phosphorylation of cyclin D1 dependent on ERK1/2, JNK, and GSK3β (proteins that are involved in cellular processes, including apoptosis) [30]. Ren et al. have proposed the detoxification effect of the combination of the two diterpenoids, cafestol and kahweol, to be the main origin of their antitumor effect. That is, the combination has been proven to down-regulate the CYP450s expression, which generally holds a pivotal role in carcinogen activation. The study also cited other previous research affirming the compounds’ capacity of inducing glutathione-S-transferase activation and increasing other phase II metabolizing enzymes. The two compounds were also reported to induce apoptosis by inhibiting certain anti-apoptotic genes and inhibit angiogenesis by modulating VEGFR2, a primary receptor of the vascular endothelial growth factor, holding great importance in the process of angiogenesis [69].

## 4. Materials and Methods

### 4.1. Chemicals, Reagents, and Equipment

All HPLC reagents and standards were of analytical grade and were procured from the companies Sigma–Aldrich (Taufkirchen, Germany) and Decorias (Rediu, Iași, Romania). A549 (human lung adenocarcinoma), T47D-KBluc (human breast cancer), and BJ (normal human fibroblasts) cells were purchased from ATCC (Manassas, VA, USA). Roswell Park Memorial Institute (RPMI) 1640 Medium, Dulbecco’s Modified Eagle Medium (DMEM), and fetal bovine serum (FBS) were bought from Antisel (Bucharest, Romania). The 2′,7′-dichlorofluorescin diacetate (DCFH-DA), dichlorodihydrofluorescein (DCFH), dichlorofluorescein (DCF), Hanks’ Balanced Salt Solution (HBSS), hydrogen peroxide (H_2_O_2_), and N-acetyl cysteine (NAC) were acquired from Sigma-Aldrich (Schnelldorf, Germany). Dimethyl sulfoxide (DMSO), phosphate-buffered saline solution (PBS), CMC, cyclophosphamide, xylazine, and ketamine hydrochloride were acquired from Sigma-Aldrich (Taufkirchen, Germany).

The HPLC system used was an 1100 series Agilent Technologies model (Darmstadt, Germany) consisting of a G1312A binary pump, an in-line G1379A degasser, a G1329A autosampler, a G1316A column thermostat, and an Agilent ion trap detector 1100 SL. Finally, a Biotek Synergy 2 multi-mode plate reader (The Lab World Group, Hudson, MA, USA) was employed for the in vitro assessments.

### 4.2. Plant Extracts

A total of 11 plant extracts were used. These were selected as a result of previous studies that aimed to identify the optimal extraction method and extraction parameters of bioactive compounds from the three plant species mentioned earlier: *Xanthium spinosum* L., *Trifolium pratense* L., and *Coffea arabica* L. The methods of extraction that were previously studied were maceration, Soxhlet extraction (SE), pertaining to the classical extraction methods, turboextraction (TBE), and ultrasound-assisted extraction (UAE), of the innovative extraction methods. The previous studies also aimed to establish a combined extraction method between TBE and UAE–UTE [31,32,33]. The used solvent was 70% ethanol and the sample-to-solvent ratio was 1:10 (*w*/*v*).

### 4.3. HPLC-MS Analysis of the Plant Extracts

The details regarding the methods and conditions used for the HPLC-MS analysis as well as the resulted chromatograms of the compounds from the resulted plant extracts can be found in the Appendix A accompanying the article [105,106,107,108,109].

### 4.4. In Vitro Assessment of Antioxidant and Anticancerous Activities

As stated before, two samples from each species were selected for further determination of in vitro biological activities, based on the determined phytochemical profile of the extracts.

#### 4.4.1. Cell Culture

For the in vitro biological activity evaluation, two malignant cell lines—A549 (human lung adenocarcinoma) and T47D-KBluc (human breast cancer)—and the normal human dermal BJ fibroblasts were used. The cells were bought from ATCC (Manassas, Virginia, USA), and grown in a humid incubator with 5% CO_2_ supplementation. T47D-KBluc cells were kept in Roswell Park Memorial Institute (RPMI) 1640 Medium, whereas BJ and A549 cells were grown in Dulbecco’s Modified Eagle Medium (DMEM) with low-glucose and high glucose. All media was supplemented with 10% Fetal Bovine Serum (FBS). The media was changed every two days, and the cells were subcultured when they had 70–80% confluence.

#### 4.4.2. Preparation of Extracts Solutions

A 250 mg/mL stock solution in dimethyl sulfoxide (DMSO) was made from lyophilized extracts. The stock solution was further diluted in DMSO to obtain working solutions. The required concentrations in the cell culture medium, which ranged from 50 to 1000 µg/mL, were achieved using these working solutions.

#### 4.4.3. Cytotoxicity—Alamar Blue Assay

The anticancerous potential of the extracts was assessed as previously described [110,111]. All three types of cells were seeded in 96-well plates and were left for 24 h to attach to the plastic substrate. The wells were washed with PBS (phosphate-buffered saline solution), and the remaining living cells were then exposed for an additional 24 or 48 h to the six selected extracts at concentrations ranging from 50 to 1000 µg/mL. The Alamar Blue (AB) assay was used to assess the ability of exposed cells to metabolically convert the non-fluorescent resazurin into the fluorescent product resorufin. Briefly, cells were washed with PBS and further incubated for 3 h with a 200 µM resazurin solution. Resorufin fluorescence was measured using a Synergy 2 multi-mode microplate reader at λ_excitation_ = 530/25 and λ_emission_ = 590/35. The experiments were conducted on three biological replicates, each one including six technical replicates. The data are expressed as relative values in comparison to the negative control (NC) (100%).

#### 4.4.4. Antioxidant Activities

The antioxidant properties of the extracts were evaluated in normal fibroblast cells using the reactive oxygen species (ROS) sensitive 2,7 dichloro-fluorescein diacetate (DCFH-DA) dye as previously described [112,113]. Following cellular internalization, intracellular esterases hydrolyze the non-fluorescent DCFH-DA dye to dichlorodihydrofluorescein (DCFH), which is contained inside the cellular compartment. In the presence of ROS, DCFH is further oxidized to dichlorofluorescein (DCF) which is proportional to the amount of intracellular ROS and can be quantified. Briefly, cells were pre-treated for 24 h with non-cytotoxic doses of the selected plant extracts and were incubated for 2 h with 100 µL of a 50 µM DCFH-DA solution prepared in Hanks’ Balanced Salt Solution (HBSS). After cell loading, non-internalized DCFH-DA was washed, and the cells were incubated for 2 h with 250 µM H_2_O_2_ in HBSS or solely HBSS to quantify ROS in stimulated and non-stimulated conditions. The fluorescence was measured using Synergy 2 multi-mode microplate reader at λ_excitation_ = 485/20 and λ_emission_ = 528/20. The potency of the XS extracts was compared to N-acetyl cysteine (NAC) treatment (20 mM solution). The experiments were conducted on three biological replicates, each one including six technical replicates. The data are expressed as relative values in comparison to the negative control (NC) (100%).

### 4.5. In Vivo Assessment of the Antitumor Activity

#### 4.5.1. Tumor Cell Viability

For this section of the study, only three extracts were used, one SE extract per plant. Sixty Swiss albino male mice with Ehrlich ascites carcinoma (EAC), 8 weeks old, and 30 g weight were used for the in vivo study of the antitumor activity of the extracts. The animals were acquired from the animal department of “Iuliu Hațieganu” University of Medicine and Pharmacy Cluj-Napoca. The animal subjects were left to acclimatize for 24 h in the vivarium of the Physiology Department.

Experiments were approved by the Ethics Committee Board of “Iuliu Hațieganu” University of Medicine and Pharmacy, Cluj-Napoca, Romania (291/23.02.2022), on Animal Welfare in accordance with the Directive 2010/63/EU, regarding the protection of animals used for scientific purposes.

The mice were randomly divided into eight groups (*n* = 7). An aliquot of 1 mL of ascites liquid with 1 million cells from a donor animal was inoculated into the mice used in the experiment. After 24 h, the mice were enrolled in the experiment and treated by oral gavage for 10 days, as follows: group 1–0.5 mL 2% CMC (positive control group—CMC); group 2–25 mg/b.w. cyclophosphamide (CYC) administered intraperitoneally (negative control group); group 3–20 mg/b.w. XS in 0.2 mL CMC administered by gavage; group 4–20 mg/b.w. TS in 0.2 mL CMC administered by gavage; group 5–20 mg/b.w. CS in 0.2 mL CMC administered by gavage; group 6–20 mg/b.w. XS in 0.2 mL CMC administered by gavage and 25 mg/b.w. cyclophosphamide administered intraperitoneally; group 7–20 mg/b.w. TS in 0.2 mL CMC administered by gavage and 25 mg/b.w. cyclophosphamide administered intraperitoneally; and group 8–20 mg/b.w. CS in 0.2 mL CMC administered by gavage and 25 mg/b.w. cyclophosphamide administered intraperitoneally.

The animals were monitored for a period of 10 days. Under anesthesia induced by an intraperitoneal injection of 90 mg/kg ketamine and 10 mg/kg xylazine, at 24 h after the last treatment. Ascites volume, liver tissue, and blood samples were collected. Samples were stored at −80 °C until further analysis. The following biological parameters were evaluated: tumor cell viability and oxidative stress markers were also performed.

#### 4.5.2. Evaluation of Stress Markers from Ascites and Liver Samples

The oxidative stress parameters were evaluated in ascites samples and in liver homogenates by quantification of malondialdehyde (MDA), a parameter of lipid peroxidation [114], and by measurement of glutathione-reduced (GSH) and oxidized (GSSG) levels and their ratio (GSH/GSSG) [115]. Additionally, catalase (CAT) and glutathione peroxidase (GPx) activities were evaluated from both tissues [116,117]. For oxidative stress assessment, the liver fragments were harvested and homogenized with a polytron homogenizer (Brinkman Kinematica, Switzerland) as previously published [118], and then a cytosolic fraction was prepared [118]. The protein level in liver tissue homogenates and in ascites samples were measured by a method described by Bradford [119].

### 4.6. Statistical Analysis of the In Vitro and In Vivo Assessments

Regarding the in vitro studies, all samples were analyzed in triplicate (*n* = 3) and the data were reported as the mean ± standard deviation (SD). The data were statistically analyzed by one-way analysis of variance (ANOVA) with a post hoc Holm-Sidak test using SigmaPlot 11.0 computer software (Systat, Software Inc., Chicago, IL, USA). Differences showing a *p*-level of 0.05 or lower were considered statistically significant.

Data analysis for the in vivo studies was also performed by one-way ANOVA and Tukey’s multiple comparisons post-test. GraphPad Prism 8 was used as software. A *p*-value < 0.05 was considered statistically significant. The results were expressed as mean ± standard deviation. The normality of distribution was tested with Kolmogorov–Smirnov test, a nonparametric test.

## 5. Conclusions

The antitumor activities of three separate plant species were analyzed: *Xanthium spinosum* L., *Trifolium pratense* L., and *Coffea arabica* L., namely green coffee beans. All extracts displayed higher cytotoxicity towards the tumor cell phenotypes than for the normal cell population. From the evaluated extracts, the *Xanthium spinosum* L. extracts showed the highest antitumoral effects, on both tumor cell lines and the highest antioxidant potential on the normal cells. All extracts induced in the normal cells a hormetic response at the intermediary doses effect associated with the presence of bioactive compounds that stimulate the cellular metabolism. In terms of cell viability, the most significant results were achieved by CS, that is, green coffee beans SE extract, followed by TS, that is, *Trifolium pratense* L. SE extract. In vivo study, ascites cell viability decreased after *T. pratense* L. and green coffee bean extract administration, whereas the oxidative stress was significant in tumor samples after *T. pratense* L. or CYC associated with *X. spinosum* L. and *T. pratense* L. extracts. Additionally, CYC associated with CS extract improved the activities of enzymatical antioxidants in ascites, whereas in liver samples, only CS and TS increased the CAT activities These results suggested that *T. pratense* L. or *X. spinosum* L. extracts in combination with chemotherapy can induce lipid peroxidation in tumor cells and decreased the tumor viability, especially *T. pratense* L. extract.

## Figures and Tables

**Figure 1 plants-12-01840-f001:**
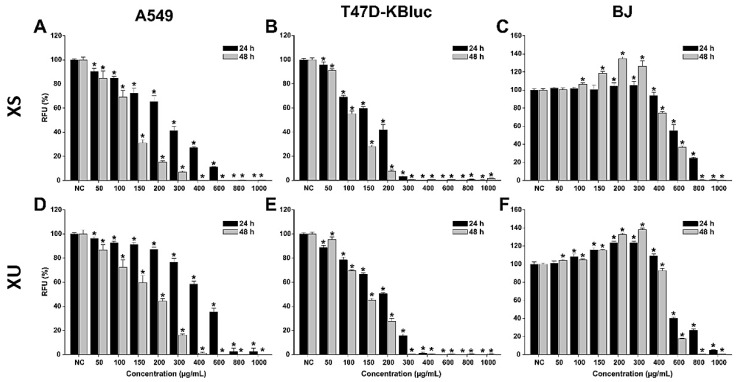
Cytotoxic effects of the XS (**A**–**C**) and XU (**D**–**F**) XU extracts observed on A549 (**A**,**D**), T47D-KBluc (**B**,**E**), and BJ (**C**,**F**) cells using the Alamar Blue Assay. The results are presented as relative means ± standard deviations of three biological replicates, where the negative control (NC) is 100%. Asterisks (*) indicate a statistically significant difference (*p* < 0.05) in comparison with the NC. RFU—relative fluorescence units.

**Figure 2 plants-12-01840-f002:**
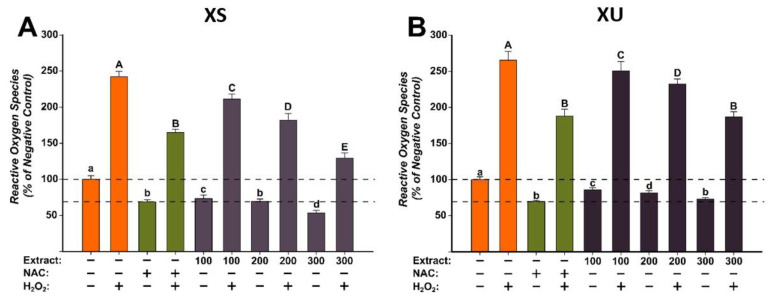
The antioxidant effect of the *X. spinosum* L. extracts was evaluated using DCFH-DA assay on BJ cells: (**A**)—*Xanthium spinosum* L. Soxhlet extract (XS); (**B**)—*Xanthium spinosum* L. UAE extract (XU). Cells were pre-exposed to the extracts (100, 200, and 300 µg/mL) or NAC (20 mM) for 24 h, and further incubated with 50 µM DCFH-DA. The antioxidant effect of the XS extracts was evaluated after 2 h in non-stimulated (HBSS) and stimulated conditions (250 µM H_2_O_2_). The data are expressed as relative means ± standard deviations (six technical replicates for each of the three biological replicates) where the negative control is 100%. Different letters (a–d refers to comparisons on non-stimulated conditions, whereas A–E refers to comparisons in stimulated conditions) indicate significant differences (ANOVA + Holm-Sidak post hoc, *p* < 0.05).

**Figure 3 plants-12-01840-f003:**
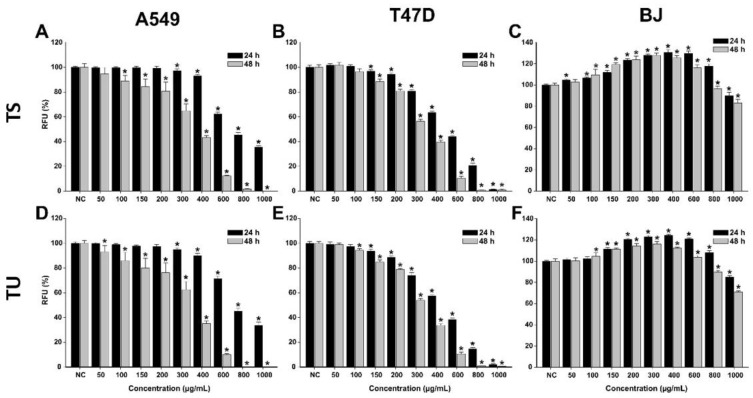
Cytotoxic effect of the TS (**A**–**C**) and TU (**D**–**F**) observed using Alamar Blue assay on A549 (**A**,**D**), T47D-KBluc (**B**,**E**), and BJ (**C**,**F**) cells. The results are expressed as relative means ± standard deviations (six technical replicates for each of the three biological replicates), where the negative control (DMSO 0.2%) is 100%. Asterisks (*) indicate a significant difference (*p* < 0.05) in comparison with the negative control. RFU—relative fluorescence units.

**Figure 4 plants-12-01840-f004:**
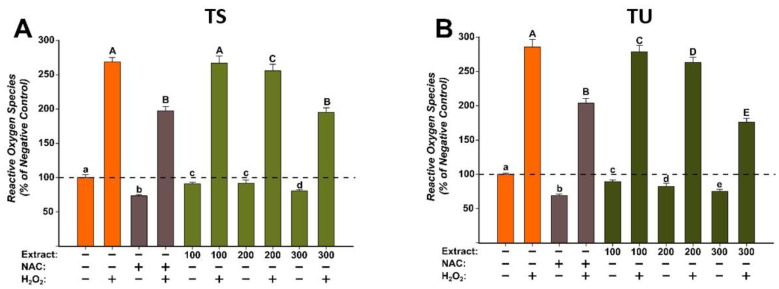
The antioxidant effect of the *T. pratense* L.: (**A**)—*Trifolium* pratense L. Soxhlet extract (TS) and (**B**)—*Trifolium pratense* L. UAE extract (TU) was evaluated using DCFH-DA assay on BJ cells. Cells were pre-exposed to the extracts (100, 200, and 300 µg/mL) or NAC (20 mM) for 24 h, and further incubated with 50 µM DCFH-DA. The antioxidant effect of the *T. pratense* L. extracts was evaluated after 2 h in non-stimulated (HBSS) and stimulated conditions (250 µM H_2_O_2_). The data are expressed as relative means ± standard deviations (6 technical replicates for each of the three biological replicates) where the negative control is 100%. Different letters (a–e refers to comparisons on non-stimulated conditions, whereas A–E refers to comparisons in stimulated conditions) indicate significant differences (ANOVA + Holm-Sidak post hoc, *p* < 0.05).

**Figure 5 plants-12-01840-f005:**
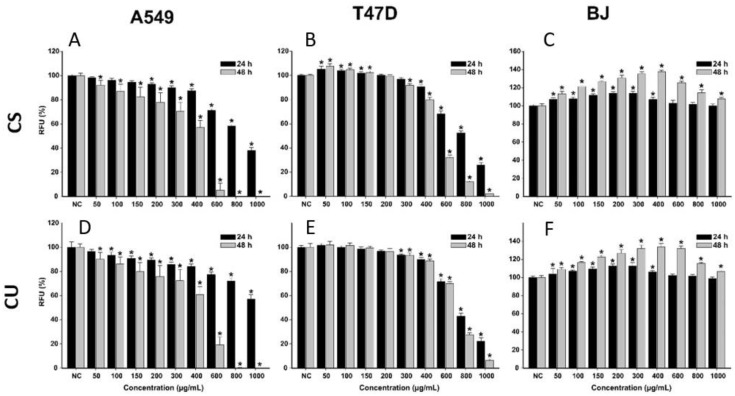
Cytotoxic effect of the CS (**A**–**C**) and CU (**D**–**F**) on A549 (**A**,**D**), T47D-KBluc (**B**,**E**), and BJ (**C**,**F**) cells. Data are presented as relative means ± standard deviations (6 technical replicates for each of the three biological replicates), where the negative control (NC) is 100%. Asterisks (*) mark a significant difference (*p* < 0.05) in comparison with the NCRFU–relative fluorescence units.

**Figure 6 plants-12-01840-f006:**
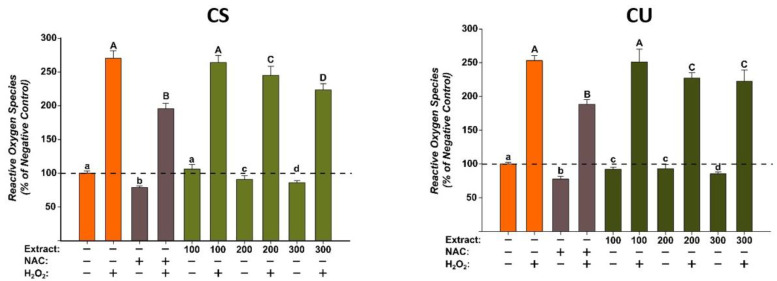
The antioxidant effect of the CS and CU extracts was evaluated using DCFH-DA assay on normal fibroblast cells (BJ). Cells pre-exposed for 24 h to the extracts (100, 200, and 300 µg/mL) or NAC (20 mM), were further incubated with 50 µM DCFH-DA. The antioxidant properties of the CA extracts were measured after 2 h in non-stimulated (HBSS) and stimulated conditions (250 µM H_2_O_2_). The data are presented as relative means ± standard deviations of three biological replicates (six technical replicates/biological replicate) in comparison with the negative control (100%). Different letters (non-stimulated conditions (a–d) and stimulated conditions (A–D)) mark significant differences (ANOVA + Holm-Sidak post hoc, *p* < 0.05).

**Figure 7 plants-12-01840-f007:**
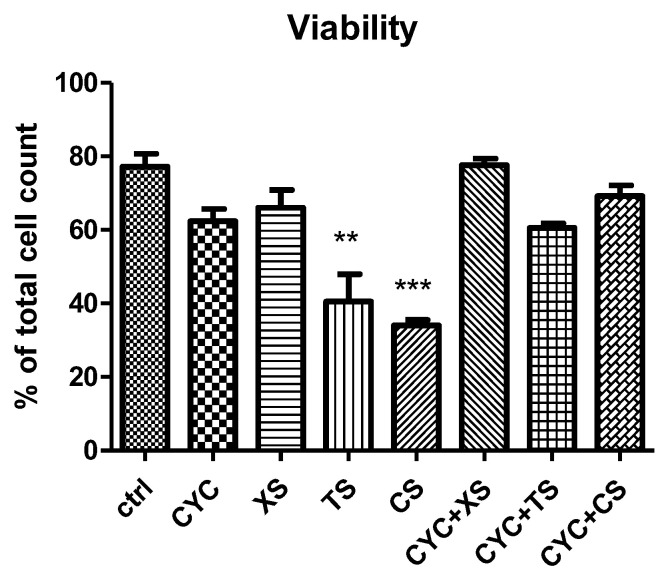
Cell viability. Values are means ± SD. Statistical analysis was performed using a one-way ANOVA, with Tukey’s multiple comparisons post-test (** *p* < 0.01 and *** *p* < 0.001 vs. control group).

**Figure 8 plants-12-01840-f008:**
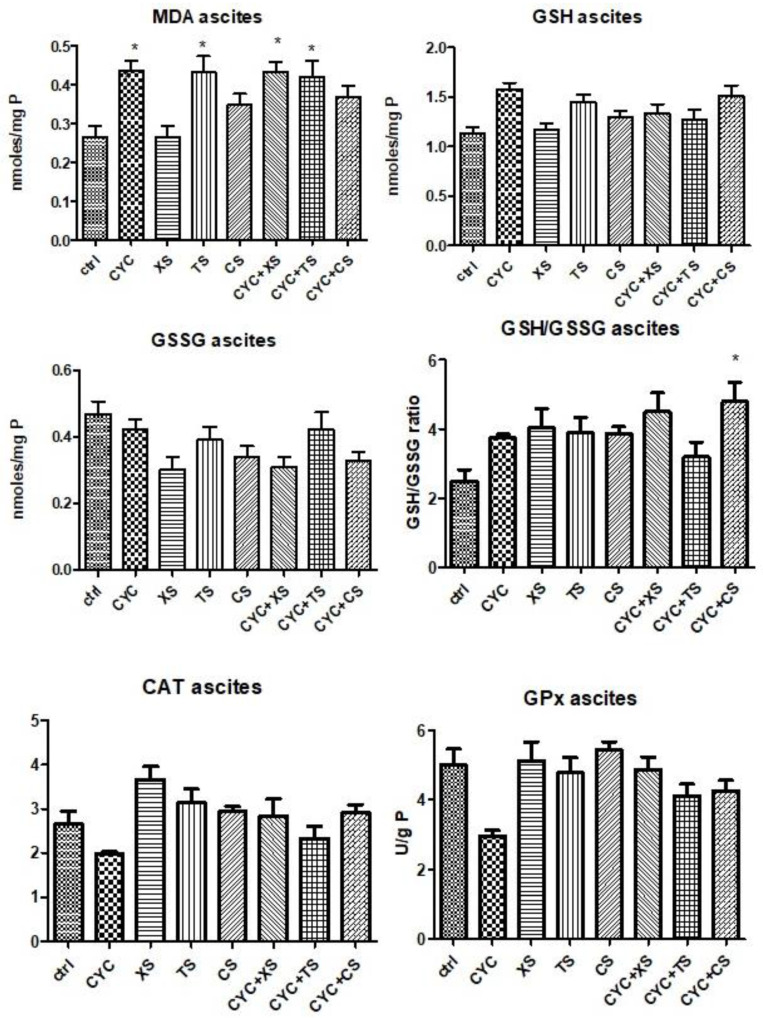
Oxidative stress markers (MDA levels, GSG/GSSG ratio, CAT, and GPx activities) from the mice ascites samples after a 10-day treatment with CYC, XS, TS, CS, and associations of CYC and each of the separate extracts (Values are means ± SD. Statistical analysis was performed using a one-way ANOVA, with Tukey’s multiple comparisons post-test (* *p* < 0.05 vs. control group).

**Figure 9 plants-12-01840-f009:**
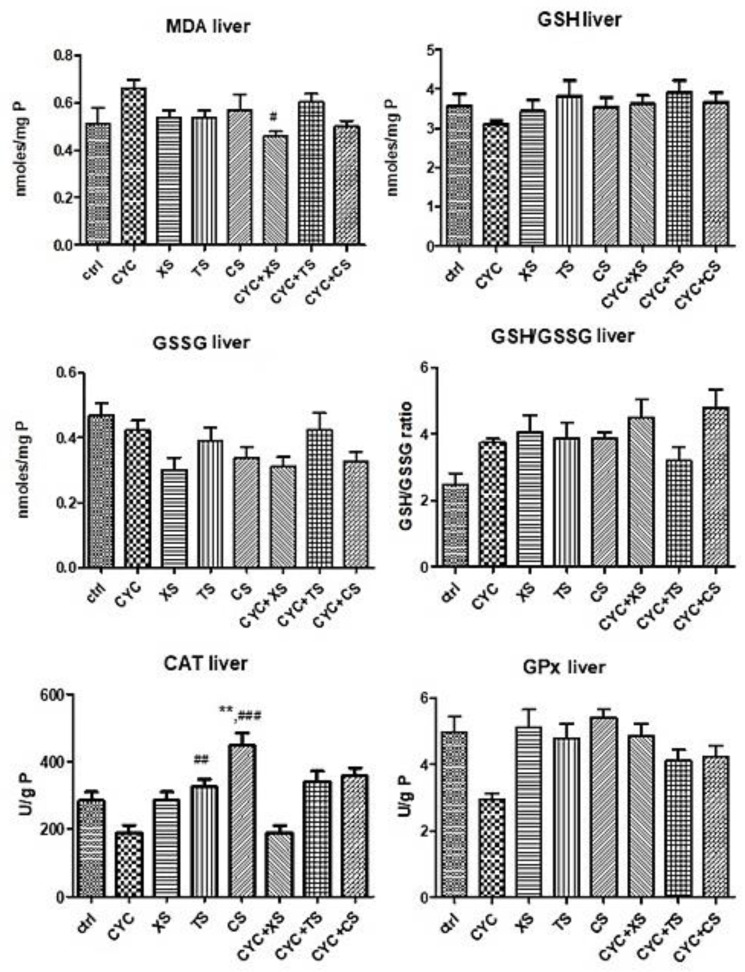
Oxidative stress markers (MDA levels, GSG/GSSG ratio, CAT, and GPx activities) from the mice liver samples after a 10-day treatment with CYC, XS, TS, CS, and associations of CYC and each of the separate extracts (Values are means ± SD. Statistical analysis was performed using a one-way ANOVA, with Tukey’s multiple comparisons post-test (** *p* < 0.01 vs. control group, and # *p* < 0.05, ## *p* < 0.01, ### *p* < 0.001 vs. CYC group).

**Table 1 plants-12-01840-t001:** Antitumor compounds specific to the genus *Xanthium*, namely the species studied, *X. spinosum* L.

	Caffeoylquinic Acids				Xanthanolide Sesquiterpene Lactones	
Sample	4-O-Caffeoylquinic Acid (µg/mL Extract) *	3,4-Dicaffeoylquinic Acid (µg/mL Extract) *	3,5-Dicaffeoylquinic Acid (µg/mL Extract) *	4,5-Dicaffeoylquinic Acid (µg/mL Extract) *	Xanthatin (µg/mL Extract) *	8-Epi-Xanthatin (µg/mL Extract) *
**M**	2.927 ± 0.351	4.09 ± 0.573	13.94 ± 0.837	21.40 ± 1.926	3.95 ± 0.513	1.04 ± 0.083
**S60**	4.611 ± 0.323	13.42 ± 0.671	56.78 ± 4.543	75.96 ± 9.875	7.22 ± 0.433	2.88 ± 0.230
**T26**	2.086 ± 0.271	7.31 ± 0.657	36.91 ± 1.845	36.94 ± 2.586	2.79 ± 0.195	1.82 ± 0.055
**T44**	2.009 ± 0.201	6.37 ± 0.574	35.48 ± 2.129	36.26 ± 2.901	3.21 ± 0.385	2.19 ± 0.132
**T46**	1.78 ± 0.267	6.58 ± 0.263	31.55 ± 1.578	35.24 ± 4.581	3.12 ± 0.405	1.49 ± 0.179
**T48**	1.244 ± 0.087	4.09 ± 0.491	17.16 ± 0.515	20.83 ± 3.125	1.73 ± 0.138	1.04 ± 0.104
**U14**	1.856 ± 0.260	10.52 ± 1.052	58.21 ± 8.149	48.17 ± 6.743	4.55 ± 0.227	6.04 ± 0.725
**U24**	2.468 ± 0.173	10.73 ± 0.429	66.90 ± 8.028	59.85 ± 1.796	5.57 ± 0.223	6.29 ± 0.503
**U25**	2.927 ± 0.234	12.49 ± 1.124	75.82 ± 10.615	72.67 ± 6.540	6.36 ± 0.191	5.79 ± 0.752
**U34**	3.234 ± 0.194	16.01 ± 1.121	106.88 ± 10.688	95.24 ± 14.286	7.27 ± 0.945	5.26 ± 0.368
**UT**	2.774 ± 0.222	13.11 ± 0.524	93.67 ± 9.367	73.92 ± 7.392	5.97 ± 0.776	4.53 ± 0.181

* concentrations were expressed as mean ± SD.

**Table 2 plants-12-01840-t002:** Antitumor compounds specific to *Trifolium pratense* L., found in the selected extracts.

Sample	Formononetin (ng/mL Extract) *	Ononin (ng/mL Extract) *	Biochanin A (µg/mL Extract) *
**M**	685.70 ± 54.856	4596.50 ± 321.755	80.48 ± 2.414
**S40**	636.43 ± 31.821	5748.17 ± 287.409	52.56 ± 4.205
**S60**	760.05 ± 68.405	6806.60 ± 1020.991	102.78 ± 11.305
**T24**	633.61 ± 57.025	4078.19 ± 122.346	73.56 ± 8.827
**T26**	581.22 ± 87.184	3157.60 ± 94.728	53.79 ± 7.530
**T46**	682.13 ± 40.928	4864.17 ± 291.850	77.40 ± 8.514
**U14**	728.94 ± 21.868	5923.07 ± 888.460	74.43 ± 8.188
**U25**	728.70 ± 65.583	5715.42 ± 685.850	95.28 ± 12.387
**U34**	675.75 ± 74.332	5254.89 ± 420.391	66.72 ± 7.339
**U35**	767.35 ± 99.755	5467.14 ± 492.042	100.81 ± 13.106
**UT**	738.34 ± 66.450	4758.77 ± 190.351	85.56 ± 12.833

* concentrations were expressed as mean ± SD.

**Table 3 plants-12-01840-t003:** Antitumor compounds characteristic of green coffee beans.

	Caffeoylquinic Acids				Diterpenes	
Sample	4-*O*-Caffeoylquinic Acid (µg/mL Extract) *	3,4-Dicaffeoylquinic Acid (µg/mL Extract) *	3,5-Dicaffeoylquinic Acid (µg/mL Extract) *	4,5-Dicaffeoylquinic Acid (µg/mL Extract) *	Kahweol (µg/mL Extract) *	Cafestol (µg/mL Extract) *
**M**	197.088 ± 19.709	80.50 ± 4.830	93.08 ± 3.723	101.82 ± 5.091	0.000 ± 0.000	0.000 ± 0.000
**S60**	265.507 ± 37.171	103.51 ± 8.281	145.08 ± 21.762	127.00 ± 10.160	1.206 ± 0.084	0.576 ± 0.052
**T26**	278.671 ± 19.507	111.80 ± 13.417	150.31 ± 13.528	139.14 ± 6.957	2.644 ± 0.397	2.482 ± 0.323
**T44**	337.217 ± 23.605	139.38 ± 15.332	187.56 ± 11.254	177.02 ± 21.243	1.874 ± 0.281	1.490 ± 0.224
**T46**	217.828 ± 8.713	86.92 ± 11.300	107.12 ± 10.712	100.69 ± 8.054	0.000 ± 0.000	0.000 ± 0.000
**T48**	269.181 ± 40.377	124.04 ± 4.962	179.11 ± 16.120	151.96 ± 18.235	10.298 ± 1.133	11.062 ± 1.549
**U14**	283.339 ± 17.000	133.26 ± 10.661	191.37 ± 28.705	162.62 ± 17.888	10.694 ± 0.642	10.916 ± 0.764
**U24**	181.476 ± 19.962	76.04 ± 2.281	106.88 ± 8.550	93.99 ± 12.219	9.992 ± 0.799	10.968 ± 0.658
**U25**	216.298 ± 10.815	92.52 ± 6.477	133.06 ± 11.975	113.50 ± 5.675	11.751 ± 0.823	12.881 ± 1.546
**U34**	255.941 ± 30.713	110.04 ± 5.502	156.15 ± 17.176	135.17 ± 5.407	9.953 ± 1.294	10.008 ± 1.301
**UT**	202.063 ± 30.309	90.14 ± 9.014	129.61 ± 16.849	113.96 ± 13.675	4.346 ± 0.174	3.881 ± 0.388

* concentrations were expressed as mean ± SD.

**Table 4 plants-12-01840-t004:** Nomenclature of the samples.

Plant Species	Selected Extracts for Biological Activity Assessment		Renamed Extracts	
*Xanthium spinosum* L.	S60 (Soxhlet extraction, 60 min extraction time)	U34 (UAE at 30 °C extraction temperature, and 40 min extraction time)	XS	XU
*Trifolium pratense* L.	S60 (Soxhlet extraction, 60 min extraction time)	U35 (UAE at 30 °C extraction temperature, 50 min extraction time)	TS	TU
*Coffea arabica* L. (green coffee beans)	S60 (Soxhlet extraction, 60 min extraction time)	U35 (UAE at 30 °C extraction temperature, 50 min extraction time).	CS	CU

**Table 5 plants-12-01840-t005:** IC_50_ values (µg/mL) after exposure of A549, T47D–Kbluc, and BJ to the XS and XU extracts for 24 h.

Sample	IC_50_ (µg/mL)					
	24 h			48 h		
	A549	T47D–KBluc	BJ	A549	T47D–KBluc	BJ
**XS**	257.31	160.61	637.81	122.16	107.36	520.38
**XU**	460.54	189.08	576.35	169.10	137.45	497.68

**Table 6 plants-12-01840-t006:** IC_50_ values (µg/mL) after exposure of A549, T47D–Kbluc, and BJ to the TS and TU extracts for 24 h.

Sample	IC_50_ (µg/mL)					
	24 h			48 h		
	A549	T47D–KBluc	BJ	A549	T47D–KBluc	BJ
**TS**	712 (692–734)	534 (519–550)	>1000	358 (348–369)	332 (323–342)	>1000
**TU**	765 (747–786)	477 (462–493)	>1000	329 (319–340)	314 (305–323)	>1000

**Table 7 plants-12-01840-t007:** IC_50_ values (µg/mL) after exposure of A549, T47D–Kbluc, and BJ to the CS and CU extracts for 24 h.

Sample	IC_50_ (µg/mL)					
	24 h			48 h		
	A549	T47D–KBluc	BJ	A549	T47D–KBluc	BJ
**CS**	879 (823–932)	757 (748–797)	>1000	401 (382–420)	519 (497–542)	>1000
**CU**	>1000	797 (753–840)	>1000	424 (396–424)	688 (660–716)	>1000

## Data Availability

Not applicable.

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
