# Peer review of "An In Vitro and In Vivo Assessment of Antitumor Activity of Extracts Derived from Three Well-Known Plant Species"

_plants, 2023, doi:10.3390/plants12091840_

Round 1

Reviewer 1 Report

In the original paper entitled “An in vitro and in vivo assessment of antitumor activity of extracts derived from 3 well-known plant species, yet left underresearched in regard to this highly sought biological effect” by Dr Gligor et al., the Authors used in vitro studies on cancer cel lines followed by the use of ascites mouse model to analyse the effects of the extracts of

 Xanthium spinosum, Trifolium pratense, and Coffea arabica (green beans). In my opinion, the results of the study are interesting and the paper is decently written. Despite this, I have one major and several minor comments listed below. In general, a quality of presentation should be improved.

 Major comment

Sections regarding statistical analysis (2.4.5 Statistical analysis of the in vitro studies and 2.5.2. Statistical analysis of the in vivo assessments) should be combined into one section. The Authors showed mean and SD and used parametric tests such as ANOVA followed by post-hoc tests, however to do this, data should be normally distributed. Did the Authors verified a normality of distribution? Please, give this information (and name of the test) in the description of statistical analysis.

Minor comments

1.    In the title, please write in full: 3 -> three. Also, please consider rephrasing the title since the style is rather awkward.

2.    Abstract should be shortened since it exceeds a length typical for original papers. The limit of 250 words should applied.

3.    Lines 120-121: “The following equipment were used: 204 Sigma Centrifuge (Osterode am Harz, Germany); Analytical Plus Balance (Mettler-Toledo, Switzerland); Ultrasonic bath Elma Transsonic 700/H (Singen, Germany).” I do not think that a description of this very typical laboratory equipment is necessary – please delete.

4.    Conclusions should state the most important facts and implications from the study. The authors instead, decided to prepare another version of the abstract (see e.g. the lines 925-934). This section should be shortened and vastly rephrased.

I do not have specific comments regarding a quality of English.

Author Response

An in vitro and in vivo assessment of antitumor activity of extracts derived from three well-known plant species

(Previously titled: “An in vitro and in vivo assessment of antitumor activity of extracts derived from 3 well-known plant species, yet left under-researched in regard to this highly sought biological effect”)

Octavia Gligor, Simona Clichici, Remus Moldovan, Nicoleta Decea, Ana-Maria Vlase, Ionel Fizeșan, Anca Pop, Piroska Virag, Gabriela Adriana Filip, Laurian Vlase, Gianina Crișan

Esteemed Editor and Reviewers,

The authors of this paper wish to express their gratitude for your feedback. Please find below the authors’ comments on the current review report.

Reviewer #1 – Comments and replies

Point

Editor comments

Authors’ comments

1

Sections regarding statistical analysis (2.4.5 Statistical analysis of the in vitro studies and 2.5.2. Statistical analysis of the in vivo assessments) should be combined into one section. The Authors showed mean and SD and used parametric tests such as ANOVA followed by post-hoc tests, however to do this, data should be normally distributed. Did the Authors verified a normality of distribution? Please, give this information (and name of the test) in the description of statistical analysis.

For statistical analysis for in vivo study, one way ANOVA and posttest Tukey was applied. The normality of distribution was tested with Kolmogorov – Smirnov test, nonparametric test .

2

In the title, please write in full: 3 -> three. Also, please consider rephrasing the title since the style is rather awkward

The modification was added.

3

Abstract should be shortened since it exceeds a length typical for original papers. The limit of 250 words should applied.

The Abstract was shortened to best fit the word limit and to also meet the requirements of the other reviewer.

4

Lines 120-121: “The following equipment were used: 204 Sigma Centrifuge (Osterode am Harz, Germany); Analytical Plus Balance (Mettler-Toledo, Switzerland); Ultrasonic bath Elma Transsonic 700/H (Singen, Germany).” I do not think that a description of this very typical laboratory equipment is necessary – please delete.

These lines were deleted as requested.

5

Conclusions should state the most important facts and implications from the study. The authors instead, decided to prepare another version of the abstract (see e.g. the lines 925-934). This section should be shortened and vastly rephrased.

The Conclusions section was shortened and rephrased accordingly.

Reviewer 2 Report

Review comments to the author

Title: ''An in vitro and in vivo assessment of antitumor activity of extracts derived from 3 well-known plant species, yet left under researched in regard to this highly sought biological effect''.

Manuscript ID: plants-2349954.

Abstract:

- The most promising biological results should be mentioned.

Keywords:

- The scientific names of species should be written in italic font.

2. Materials and Methods:

2.3. HPLC-MS analysis of the plant extracts

- The term ''m/z'' should be written in italic font.

- This section should be moved to the supplementary file.

2.4. In vitro assessment of antioxidant and anticancerous activities

2.4.5. Statistical analysis of the in vitro studies

- In this subtitle, the word ''in vitro'' should be written in italic font.

2.5. In vivo assessment of the antitumor activity

2.5.2. Statistical analysis of the in vivo assessments

- In this subtitle, the word ''in vivo'' should be written in italic font.

3. Results:

3.1. HPLC-MS analysis of the extracts

- Insert HPLC-MS chromatograms.

5. Conclusions:

- Conclusion part should be more concise and supported by the results.

Citations in the text:

- Text citation should be written between closed brackets [] not Parentheses ().

Abbreviations:

- A list of abbreviations should be inserted by the end of the manuscript before references.

Author Response

An in vitro and in vivo assessment of antitumor activity of extracts derived from three well-known plant species

(Previously titled: “An in vitro and in vivo assessment of antitumor activity of extracts derived from 3 well-known plant species, yet left under-researched in regard to this highly sought biological effect”)

Octavia Gligor, Simona Clichici, Remus Moldovan, Nicoleta Decea, Ana-Maria Vlase, Ionel Fizeșan, Anca Pop, Piroska Virag, Gabriela Adriana Filip, Laurian Vlase, Gianina Crișan

Esteemed Editor and Reviewers,

The authors of this paper wish to express their gratitude for your feedback. Please find below the authors’ comments on the current review report.

Reviewer #2 – Comments and replies

Point

Editor comments

Authors’ comments

1

Abstract:

- The most promising biological results should be mentioned.

The most important results were added.

2

Keywords:

- The scientific names of species should be written in italic font.

The scientific names of the species have been rewritten in italic font.

3

2. Materials and Methods:

2.3. HPLC-MS analysis of the plant extracts

- The term ''m/z'' should be written in italic font.

The term “m/”z was rewritten in italic font throughout all the manuscript.

4

- This section should be moved to the supplementary file.

A supplementary material has been created as requested.

5

2.4. In vitro assessment of antioxidant and anticancerous activities

2.4.5. Statistical analysis of the in vitro studies

- In this subtitle, the word ''in vitro'' should be written in italic font.

The phrase “in vitro” was rewritten in italic font.

6

2.5. In vivo assessment of the antitumor activity

2.5.2. Statistical analysis of the in vivo assessments

- In this subtitle, the word ''in vivo'' should be written in italic font.

The phrase “in vivo” was also rewritten in italic font.

7

3. Results:

3.1. HPLC-MS analysis of the extracts

- Insert HPLC-MS chromatograms

Examples of the HPLC chromatograms were included as requested in the Supplementary Material, now accompanying the manuscript.

8

5. Conclusions:

- Conclusion part should be more concise and supported by the results.

The Conclusion section was rewritten as you suggested.

9

Citations in the text:

- Text citation should be written between closed brackets [] not Parentheses ().

The citations were rewritten throughout the entire manuscript in closed brackets [].

10

Abbreviations:

- A list of abbreviations should be inserted by the end of the manuscript before references.

The list of Abbreviations was included at the end of the manuscript before references, as requested.